# A Mix of Dietary Fibres Changes Interorgan Nutrients Exchanges and Muscle-Adipose Energy Handling in Overfed Mini-Pigs

**DOI:** 10.3390/nu13124202

**Published:** 2021-11-23

**Authors:** Ahmed Ben Mohamed, Didier Rémond, Andreu Gual-Grau, Annick Bernalier-Donnadille, Frédéric Capel, Marie-Caroline Michalski, Fabienne Laugerette, Benoit Cohade, Noureddine Hafnaoui, Daniel Béchet, Cécile Coudy-Gandilhon, Marine Gueugneau, Jerome Salles, Carole Migné, Dominique Dardevet, Jérémie David, Sergio Polakof, Isabelle Savary-Auzeloux

**Affiliations:** 1Unité de Nutrition Humaine (UNH), Unité Mixte de Recherches 1019, Université Clermont Auvergne, Institut National de Recherche pour l’Agriculture, l’Alimentation et l’Environnement (INRAE), 63000 Clermont-Ferrand, France; mohamed.ahmed.ben34@gmail.com (A.B.M.); didier.remond@inrae.fr (D.R.); andreu.gual-grau@inrae.fr (A.G.-G.); frederic.capel@inrae.fr (F.C.); benoit.cohade@inrae.fr (B.C.); noureddine.hafnaoui@inrae.fr (N.H.); daniel.bechet@inrae.fr (D.B.); cecile.coudy-gandilhon@inrae.fr (C.C.-G.); Marine.Gueugneau@inrae.fr (M.G.); jerome.salles@inrae.fr (J.S.); dominique.dardevet@inrae.fr (D.D.); jeremie.david@inrae.fr (J.D.); sergio.polakof@inrae.fr (S.P.); 2Unité de Microbiologie Environnement Digestif et Santé, Unité Mixte de Recherches 0454, Université Clermont Auvergne, Institut National de Recherche pour l’Agriculture, l’Alimentation et l’Environnement (INRAE), 63000 Clermont-Ferrand, France; annick.bernalier@inrae.fr; 3CarMeN Laboratory, Unité Mixte de Recherches 1397, INRAE/Institut National de la Santé et de la Recherche Médicale (Inserm), Université Claude Bernard Lyon 1, Institut National de Recherche pour l’Agriculture, l’Alimentation et l’Environnement (INRAE), 69310 Pierre-Bénite, France; Marie-Caroline.Michalski@inrae.fr (M.-C.M.); fabienne.laugerette@univ-lyon1.fr (F.L.); 4MetaboHUB Clermont, Plateforme d’Exploration du Métabolisme, Unité de Nutrition Humaine (UNH), Institut National de Recherche pour l’Agriculture, l’Alimentation et l’Environnement (INRAE), Université Clermont Auvergne, 63000 Clermont-Ferrand, France; carole.migne@inrae.fr

**Keywords:** overfeeding, dietary fibres, gene expression, SCFA, muscle metabolism, inter-organ metabolism, obesity

## Abstract

This study evaluates the capacity of a bread enriched with fermentable dietary fibres to modulate the metabolism and nutrients handling between tissues, gut and peripheral, in a context of overfeeding. Net fluxes of glucose, lactate, urea, short chain fatty acids (SCFA), and amino acids were recorded in control and overfed female mini-pigs supplemented or not with fibre-enriched bread. SCFA in fecal water and gene expressions, but not protein levels or metabolic fluxes, were measured in muscle, adipose tissue, and intestine. Fibre supplementation increased the potential for fatty acid oxidation and mitochondrial activity in muscle (*acox*, *ucp2*, *sdha* and *cpt1-m*, *p* < 0.05) as well as main regulatory transcription factors of metabolic activity such as *pparα*, *pgc-1α* and *nrf2*. All these features were associated with a reduced muscle fibre cross sectional area, resembling to controls (i.e., lean phenotype). SCFA may be direct inducers of these cross-talk alterations, as their feces content (+52%, *p* = 0.05) was increased in fibre-supplemented mini-pigs. The SCFA effects could be mediated at the gut level by an increased production of incretins (increased *gcg* mRNA, *p* < 0.05) and an up-regulation of SCFA receptors (increased *gpr41* mRNA, *p* < 0.01). Hence, consumption of supplemented bread with fermentable fibres can be an appropriate strategy to activate muscle energy catabolism and limit the establishment of an obese phenotype.

## 1. Introduction

In order to face the rapid increase of obesity and related diseases both in Europe and the US [1,2], the development of safe and efficient strategies capable of limiting weight gain and improving health status are in high demand. Among those, the increase in dietary fibre intake has been proven to be efficient in limiting weight gain, improving insulin sensitivity, maintaining metabolic homeostasis [3,4] and correlated to a decreased risk of diabetes, cardiovascular diseases and even certain cancers [5,6]. However, less is known about these health benefits in a situation when the energy requirements are more than substantially exceeded. Yet, such situations occur frequently in obese or overfed populations when offered various naturally fibre-rich or fibre-enriched foods, two strategies regularly promoted by public health policies [7,8].

The supplementation of dietary fibres in high fat/high sucrose or in over-fed rats has been proven to limit the deleterious diet-induced effects on body weight gain, insulin resistance, and microbiota composition, as well as gut and hepatic metabolism [9,10,11]. Such a similar effect has been shown in a pig model that is closer to humans in terms of nutritional habits, microbiota composition and digestive physiology/metabolism [12]. Our group recently showed that a mix of dietary fibres (resistant starch, pectin and inulin) was capable of limiting body weight gain and the hepatic accumulation of lipid droplets in overfed mini-pigs, probably via an alteration of PPAR-regulated lipids entry into the liver [13]. A significant part of the health-promoting effect of fibres should involve their fermentation products by gut microbiota such as short chain fatty acids (SCFAs) [14], particularly propionate and butyrate [15,16]. Indeed, SCFAs are known to present pleiotropic effects. At the splanchnic level, they impact on gut inflammation and gut incretin release [17,18], hepatic energy storage (as glycogen and lipid droplets) and utilization [13,19,20]. At the peripheral level (muscle and adipose tissues), few studies previously showed that (and how) SCFAs might modulate energy expenditure and storage [21,22].

Taken together, these data suggest that in a situation of over-nutrition or higher energy supply, the beneficial effects of dietary fibres (and/or their fermentation-derived products) at the whole body level are the result of tissues-specific metabolic adaptations and shifts in energy use vs. storage, with notably a direct impact on the splanchnic tissues, namely gut/microbiota and liver [13,23]. Meanwhile, peripheral tissues such as muscle and adipose tissues also adapt their metabolism after fibre supplementation and respond to signals which remain poorly explored and may involve directly or indirectly SCFAs [21,24,25]. Additionally, one cannot exclude digestive interactions between ingested fibres and lipids, known to reduce lipid absorption and play on lipids’ endogenous secretions that could also explain part of the beneficial effects [26,27].

In this context, the aim of the present study is dual. Firstly, explore how dietary fibre supplementation impacts on energy distribution between the splanchnic area and peripheral tissues in a situation of over-nutrition. Secondly, explore which metabolic pathways are regulated by fibre supplementation in the peripheral tissues (muscle and adipose). For these purposes, the nutrients net fluxes throughout the splanchnic tissues (gut and liver) and the gene expressions of major metabolic pathways involved in the utilization and storage of nutrients in the gut, muscle and adipose tissues were jointly investigated in overfed mini pigs. This work on dietary fibre supplementation in a context of over-nutrition is based on several previous studies we implemented on the same overfed Yucatan mini-pigs, whose metabolic response to diet has already been extensively unraveled [13,28,29,30] and has shown that the majority of phenotypes described are equivalent to what we can observe in humans.

## 2. Materials and Methods

### 2.1. Animals and Experimental Procedure

The study involved 20 female adult Yucatan mini-pigs with an average body weight of 34.8 ± 1.5 kg at the beginning of the experiment. The animals were housed individually in pens (1 × 1.5 m) in a ventilated room with controlled temperature (21 °C) and regular light cycle (L12:D12). Just before starting the experiment, the 20 animals were fed with 600 g/day of a standard diet (control diet (C)) (2230 kcal/kg) containing 69.2% starch, 17% protein, 3.3% fat, 5.5% cellulose, 5% ashes (Porcyprima; Sanders Centre Auvergne, Aigueperse, France) and had free access to tap water. This level of intake corresponds to the energy requirement for these pigs and corresponds to an overall supply of energy of 1300–1400 kcal/day.

After this adaptation period, the animals were divided into 3 groups. Six pigs were used as control animals (group C). The remaining 14 mini-pigs were then surgically fitted with a catheter in the abdominal aorta, portal and hepatic veins. After 3 weeks of post-surgical recovery, 7 mini-pigs received an obesogenic diet (group O) for 56 days consisting of the standard diet (800 g) enriched with fat (10% palm oil *w*/*w*) and sugar (10% sucrose *w*/*w*) (2570 kcal/kg) and 250 g of regular bread (2044 kcal/kg). The energy intake of these animals is 2640 kcal/d. The last 7 mini-pigs received the same 800 g of the obesogenic diet, but regular bread was replaced by experimental bread (1678 kcal/kg) in which 23% of the flour was substituted with a mix of dietary fibres (group O + F) as previously described [31]. The fibre mix consisted of 20% inulin, 20% pectin and 60% resistant starch (Cargill, Minnetonka, Minnesota USA). The energy intake of the O + F fed animals is 2621 kcal/d. Food intake was recorded daily, and the animals’ body weight was measured weekly as previously published [13].

After an overnight fast, blood from the portal vein, hepatic vein and abdominal aorta was sampled through the catheters on heparinized or EDTA tubes before the 1st day of overfeeding (D1) (i.e., samples obtained at D1 correspond to animals adapted to the standard diet (such as group C)), and after 14 days (D14) and 56 days (D56) of adaptation to each experimental diet (O or O + F). These time points have been chosen because they have been considered as key steps in the evolution of the phenotype of the Yucatan mini-pigs when overfed with a similar diet in a previous work [28], D14 corresponding to the early adaptation step associated with the diet change and D56 a longer adaptation equilibrium to the obesogenic diet. Fresh feces were also sampled at the same time points and stored at −80 °C for further analysis. The experimental design has been previously published in detail [13]. Blood was centrifuged at 4500× *g* for 10 min, plasma rapidly collected and stored at −80 °C until further analyses. At D56 the mini-pigs of the groups C, O and O + F were euthanized after an overnight fast by intravenous administration of Dolethal^®^ (pentobarbitone sodium 200 mg/L, Vetoquinol^®^, Lure, France). Samples of tissues (sub cutaneous adipose tissue, muscle (*Longissimus dorsi*)) were rapidly excised and freeze clamped. A sample of muscle tissue was fixed in ice isopentane and stored at −80 °C.

All procedures were in accordance with the guidelines formulated by the European Community for the use of experimental animals (L358-86/609/EEC, Council Directive, 1986) and the accreditation number for the present study is C63 34514.

For the measurement of plasma flow at D14 and D60, 1 hour before the blood samplings, a solution of 0.185 M of sodium p-aminohippurate (PAH) (pH 7.4) was injected firstly as a flooding dose (5 mL of the PAH solution), then infused in the mesenteric vein at a rate of 12 mL/h. The plasma flows in each vessel were calculated according to the Fick principle [32]. It has to be noted that plasma flows and net nutrient fluxes data were shown at D14 and D56 because the fasted state studied represented an adaptation to a diet whose amount and composition at D1 (adaptation diet) was too different from the other 2 days of sampling (obesogenic diet ± dietary fibres).

### 2.2. Analytical Procedures

Glucose, lactate, urea and triacylglycerol (TG) concentrations were enzymatically measured using commercial kits on a clinical chemistry analyser (ABX Pentra 400, Horiba Medical, Montpellier, France). Plasma lipoprotein binding protein and interleukin 6 (IL-6) levels were assayed using commercial ELISA kits (Cliniscience, Nanterre, France and Sigma Aldrich, St. Louis, MI, USA, respectively). Beta-hydroxybutyrate (BHB) and ammonia were enzymatically measured using a commercial kit (Sigma Aldrich, St. Louis, MI, USA).

For amino acids (AA) measurement, detailed procedures are described elsewhere [33]. Shortly, plasma samples were deproteinised with sulphosalicylic acid after adding norleucine (50 µL, 1.25 mmol/L) as an internal standard. The supernatant was diluted (2/3) with a lithium injection buffer (Bioritech, Guyancourt, France) containing glucosaminic acid as an injection standard and AA concentrations were determined with an AA analyzer (Hitachi L8900, Sciencetec, Villebon/Yvette, France) by ion exchange chromatography (with post–column derivatisation with ninhydrine. Essential amino acids (EAA) included histidine, isoleucine, leucine, lysine, methionine, phenylalanine, threonine, tryptophane and valine; non-essential amino acids (NEAA) included alanine, glutamate, glutamine, glycine, serine, tyrosine, cystine, citrulline, ornithine, arginine, proline, asparagine, aspartate and taurine and branched-chain amino acids (BCAA) included isoleucine, leucine and valine. Total amino acids (TAA) were the sum of IAA and NIAA.

PAH concentration in plasma was measured according to [34], 250 µL of plasma samples were deproteinized with sulphosalicylic acid, thoroughly mixed and centrifuged at 10,000× *g*, 4 °C, for 15 min. 80 µL of the supernatant was deacetylated by adding 20 µL of 5 M HCL followed by an incubation at 90 °C for 1 h [35]. Sodium nitrite (625 mg/L) was then added manually. The samples were then inserted into the automotive ABX Pentra 400 (Horiba Medical, France), which added successively ammonium sulfamate (0.64 g/L) and N-(1-Naphtyl) ethylenediamine dihydrochloride (1 mg/mL). Concentrations were then determined by comparison with PAH standard and read out at 600 nm.

SCFAs in plasma were measured using 1-(tert-butyldimethylsilyl) imidazole (MTBSTFA) derivatization and analysis by gas chromatography (GC) according to Pouteau et al., 2001 [36]. Shortly, to 500 µL of plasma was added 50 µL of a mixture of ^13^C labelled 1-^13^C-acetate (4 mM), 1-^13^C-propionate (1.5 mM), 1-^13^C-butyrate (0.6 mM) (Cortecnec, Voisins Le Bretonneux, France) and 10 µL of 37% (*v*/*v*) HCL solution. 2 mL of diethyl ether was added to plasma and the mixture was centrifuged (10 min, 2000 rpm). 50 µL of MTBSTFA (Tokyo Chemical Industry, Tokyo, Japan) was added to the supernatant for SCFA derivatization. The mixture was injected in GC–MS system 7890A (Agilent Technologies, Santa Clara, CA, USA) using the splitless mode equipped with a quadrupole detector (5975C) and autoinjector (7683). The ionisation mode was operated in electron impact (electron energy 70 eV). The GC system was fitted with a nonpolar capillary column DB-5 MS (J&W Scientific, Folsom, CA, 30 m × 0.25 mm i.d × 0.25 μm film thickness) for chromatographic separation. Quantification of the SCFA was performed using the Selected Ion Monitoring acquisition Mode by measurement of the m/z ratios of the specific ^13^C and ^12^C ions of each quantified SCFA and comparison to a standard curve: 118/117 (1-^13^C-acetate/^12^C acetate), 132/131 (1-^13^C-propionate/^12^C propionate), 146/145 (1-^13^C-butyrate/^12^C butyrate). SCFA production in feces was quantified by 1D 1H NMR in water-extracted fecal samples (560 µL) with TSPD4 as an internal standard. Fecal water was obtained as previously described [37].

Measurements on muscle fibres’ cross-sectional area (CSA) were performed on muscle 10 µm thick cross-sections obtained at −20 °C using a cryostat (HM500M Microm International, Fisher Scientific, Illkirch, France) and labeled with anti-laminin-α1 (L9393-Sigma). Observations and image acquisitions were captured with a high-resolution, cooled digital DP-72 camera coupled to a BX-51 microscope (Olympus) and Cell-D software (Olympus Soft Imaging Solutions, Münster, Germany). The images analysis was performed using the Visilog 6.9 software (Noesis, Crolles, France).

### 2.3. RNA Extraction and Quantification

Total RNA from subcutaneous adipose tissue (300 mg), *Longissimus dorsi* muscle (150 mg), proximal caecum (100 mg) and jejunum (50 mg) were isolated by using a PureLink RNA Mini Kit (Invitrogen, Thermo Fisher Scientific, Waltham, MA, USA) according to manufacturer’s protocol. Before sample processing with column RNAse-free, tissues were homogenized in 1 mL (muscle, caecum, jejunum) or 2 mL (subcutaneous adipose) of TRIzol Reagent (Ambion, Life Technologies, Carlsbad, CA, USA), then were centrifuged (10,000× *g* for 5 or 10 min at 4 °C). The upper phase was collected and added with 200 µL (muscle, caecum, jejunum) or 250 µL (subcutaneous adipose) of chloroform. After shaking vigorously for 15 s, the tubes were centrifuged (12,000× *g* for 5 min at 4 °C). The upper phase (muscle, caecum, jejunum) was carefully transferred to an RNAse-free column. The aqueous phase of the subcutaneous adipose tissue was washed again with 400 µL of chloroform before being centrifuged and transferred to an RNAse-free column (invitrogene). RNA samples purity was verified using NanoDrop spectrophotometer (ND-1000, NanoDrop Technologies Inc., Wilmington, DE, USA) and RNA (1 µg) integrity was verified on 1% agarose ethidium-bromide-stained gel. cDNA was obtained from 1 µg of RNA using the high-capacity cDNA reverse transcription kit (AB Applied Biosystems, Foster City, CA, USA), containing a mix of random primers and oligo(dT), according to the manufacturer’s instructions.

### 2.4. Quantitative Real-Time PCR (RT-qPCR) Methods and Data Analysis

cDNA was diluted 1:20 in RNAse-free water. Pre-amplification was performed using PowerUp SYBR Green Master Mix (AB Applied Biosystems, Life Technologies, Woolston Warrington, UK). Real-time quantitative PCR was carried out on a Bio-Rad CFX-96 detection system with quantitative qPCR SYBR Green reagents. Primers were designed to span intron/exon boundaries (preventing the amplification of contaminating genomic DNA) using Primer3 software (Whitehead Institute for Biomedical Research/MIT Center, Cambridge, MA, USA), while gene sequences were obtained from public databases (PubMed, Ensembl) (Table 1). PowerUp SYBR Green Master Mix (7.5 μL) was mixed with 0.3 μL of forward (10 μM) and reverse (10 μM) primer mix and 2 μL of diluted cDNA, and incubated at 95 °C for 10 min. PCR conditions were standardized to 40 cycles of: 95 °C for 15 s, 59 °C for 30 s and 65 °C for 5 s. Relative quantification of the target genes was conducted using *eef1α* housekeeping gene (caecum, jejunum). Relative quantification was made using the comparative ∆∆Ct method for muscle (using *rps9*, *tbp*, *gapdh* and *hrp1* as reporter genes) and subcutaneous adipose (using *actin*, *eef1α*, *rps9*, *tbp*, *gapdh* and *hrp1* as reporter genes).

### 2.5. Calculations

Net nutrient fluxes through the gut (viscera drained by the portal vein), the liver and total splanchnic tissues (gut + liver) were calculated based on princeps studies from Katz et al., 1969 [32]. Shortly, the net nutrient fluxes were calculated as differences between the afferent flux and the efferent flux. Consequently, a positive net flux indicates a net release whereas a negative net flux indicates a net uptake. Metabolites (MET) net gut release (or net portal appearance) was calculated as follows: ([MET]_PV_ − [MET]_A_) × PF_PV_ where PF_PV_ is the portal plasma flow, [MET]_PV_ and [MET]_A_ the metabolite concentrations in the portal vein and the artery, respectively. The net hepatic flux of metabolites was calculated as follows: ([MET]_HV_ × PF_HV_ − ([MET]_PV_ × PF_PV_ + [MET]_A_ × PF_AH_) where [MET]_HV, PV_ and _A_ is the metabolite concentrations in the hepatic vein, portal vein and artery, respectively, and PF _HV, PV_ and _A_ are the plasma flows in the hepatic vein, portal vein and artery, respectively. Lastly, the net flux of metabolites across overall splanchnic tissues was calculated as follows: ([MET]_HV_ − [MET]_A_) × PF_HV_ where PF_HV_ is the plasma flow in the hepatic vein and [MET]_HV_ and _A_ the metabolite concentrations in the hepatic vein and artery.

### 2.6. Statistics

All data are expressed as means ± SEM. Comparisons of data between D1, D14 and D56 in the fasted state for plasma metabolites concentrations and D14 and D56 for net nutrient fluxes were performed using a two way (time and diet effects) repeated measures ANOVA (SigmaPlot 12, Systat software, San Jose, CA, USA) followed by a post hoc analysis using the Tukey test.

Comparisons between diets at D56 (data obtained in tissues) in the fasted state were obtained using a one-way ANOVA (SigmaPlot 12, Systat software, San Jose, CA, USA) followed by a post hoc analysis using the Tukey test. Comparisons between diets for the muscle fibre cross-sectional area was determined by Student’s *t*-test calculated for each fibre-size range. Differences were considered significant if *p* < 0.05 and as a tendency (t) for 0.05 < *p* < 0.1.

## 3. Results

The impact of overfeeding associated or not with dietary fibre supplementation on our animals’ metabolic phenotype can be found in a paper published previously [13]. Indeed, we reported in our previous work a significant body weight gain, alterations of metabolic parameters such as fasted plasma insulin (+88%), HOMA-IR (+102%), cholesterol (+45%) and lactate (+63%) at D56 compared to D1 and increased lipids accumulation in the liver following overfeeding relative to control animals fed at a maintenance level. Dietary fibre supplementation limited body weight gain and hepatic lipids infiltration. In the present article, the impact of the dietary fibre supplementation is targeted on two peripheral tissues (muscle, adipose tissue) and the gut. Our aim is to highlight the metabolic adaptation of the peripheral tissues (and potential causes) to overfeeding associated or not with fibres, in the fasted state. For the latter purpose, we analyzed (1) the adaptation of various metabolic pathways within muscle and adipose in a situation of nutrients overflow (in presence or not of dietary fibres in the diet), and (2) the net nutrients fluxes at the gut level as this organ is an important user/emitter of various nutrients that can modulate (or be the consequence of) the adaptations of metabolic pathways in the peripheral tissues. The study of the splanchnic area can highlight some explanatory mechanisms of metabolic adaptations observed at the peripheral level.

### 3.1. Impact of Dietary Fibre Supplementation on Muscle (Longissimus dorsi) Phenotype and Gene Expression

Overnutrition induced a shift in the *Longissimus dorsi* fibre size, with an increased number of larger muscle fibres in the O group relatively to both O + F and C, especially for muscle fibres cross-sectional area (CSA) ranging from 6300–7000 to 9100–9800 µm^2^ (*p* < 0.05) (Figure 1A). Remarkably, muscle fibre size distribution was similar between C and O + F animals. Glycogen content in the muscle at D56 was significantly lower in the O group compared to C (−48%, *p* < 0.05), while fibre supplementation attenuated the dietary impact on glycogen storages (O + F not different from O and C) (Figure 1C). Fasting triglycerides content in muscle from the three groups was not significantly different at D56 (Figure 1B).

Many gene expressions measured at the fasted state concurred to show that fibre supplementation limited muscle lipid entry/storage and facilitated lipid catabolism. Indeed, several genes involved in fatty acid oxidation and mitochondrial activity were overexpressed in the O + F group relatively to both the C and O groups (+41%, +72%, +51%, +102% in O + F vs. O for *acox* (*p* = 0.08), *sdha* (*p* = 0.04), *ucp2* (*p* = 0.03) and *cpt1-m* (*p* = 0.04), respectively), suggesting an increased activity of oxidative and mitochondrial activity in the fibre-supplemented pigs (Figure 1B). For these four genes, the C and O groups were not different.The *hsl* gene regulating the degradation of triglycerides into free fatty acids was also more expressed in the fibre-supplemented overfed group than in the control and non-supplemented overfed animals (+116% in O + F vs. C, *p* = 0.02; +63% in O + F vs. O, *p* = 0.07). Moreover, the expression of *Slc27a4*, a gene related to lipids entry into the myocytes, was significantly decreased in the fibre-supplemented pigs (−57% in O + F vs. C, *p* = 0.05), whereas the O animals showed intermediate values (not significantly different from the C and O + F groups).

In adaptation to overfeeding conditions, fibre supplementation did not alter the expression of enzymes involved in de novo lipogenesis (*fasn*) compared to the O group (O + F vs. O, *p* = 0.72), whereas it was significantly increased compared to C (+195% in O + F vs. C, *p* = 0.03). However, despite that fatty acids can be synthetized in the muscle, the global contribution of de novo lipogenesis to total myocytes’ fatty acid flux can be considered as minor [38].

Hexokinase (*hk1*), involved in glucose (or other hexoses) phosphorylation (i.e., first step of glycolysis), gene expression in muscle was significantly decreased in the O + F group (−69% in O + F vs. C, *p* = 0.05) but not in the O group (−54% in O vs. C, *p* = 0.14). These genes are regulated by several proteins such as *srebp1c*, whose expression levels were similar between the C, O and O + F groups. However, the expression levels of genes encoding for other proteins involved in the regulation of energy metabolism and homeostasis were uniquely increased in the fibre-supplemented pigs: *pparα* and *pgc1-α* (+50% and +107% in O + F vs. C, *p* = 0.07 and *p* < 0.01, respectively; +46% and +58% in O + F vs. O, *p* = 0.08 and *p* = 0.02, respectively), whereas these gene expressions remained unaltered between the C and O groups. In the same way, the transcription factor *nrf2*, involved in the regulation of antioxidant response, was strongly increased by fibre supplementation (+197% in O + F vs. C, *p* < 0.01; +182% in O + F vs. O, *p* < 0.01), whereas C and O were similar. Lastly, the expression of *tnfα* and *Il-6* was not altered after any dietary treatment.

### 3.2. Impact of Dietary Fibre Supplementation on Subcutaneous Adipose Tissue, Caecum and Jejunum Genes Expression

Contrary to muscle, some but not all genes measured involved in fatty acids beta oxidation and mitochondrial activity were increased by fibre supplementation in adipose tissues (Figure 2). This was the case for *acox*, whose expression tended to increase only in the O + F group vs. C (*p* = 0.07), with O intermediate (Figure 2). Moreover, the overfeeding induced a down-regulation of *cpt1-l* gene expression (−50% in O vs. C, *p* = 0.002), which was reverted by fibre supplementation with an increased *cpt1-l* expression (+64% in O + F vs. O, *p* = 0.05), thereby restoring an expression similar to C. For mitochondrial *cpt2* and *sdha* gene expression, O and O + F were similar but up-regulated relatively to C (Diet effect, *p* = 0.004 and *p* = 0.02 for *cpt2* and *sdha*, respectively). Aside from oxidative activity, a stimulation of fatty acids synthesis potential, measured by *fasn* gene expression, was observed by overfeeding independently of dietary fibre supplementation (Diet effect, *p* = 0.001), whereas triglycerides hydrolysis potential (measured by *hsl* gene expression) was not affected by dietary intervention.

These metabolic processes are under the regulatory control of several transcription factors such as *pparα* and *srebf1c*, related to mitochondrial fatty acid β-oxidation and adipocyte lipogenesis, respectively. The gene expression of *pparα* was increased by fibre supplementation relatively to both the control and overfed animals (+94% in O + F vs. C, *p* < 0.05; +41% in O + F vs. O, *p* < 0.05). However, *srebf1c* expression levels, dependent on *pparα* regulatory control, remained unaltered in this tissue (Diet effect, *p* = 0.16). Finally, *angptl4* gene expression, involved in the regulation of glucose homeostasis, lipid metabolism and insulin sensitivity, was significantly down-regulated in the O + F group compared to controls (−50% in O + F vs. C, *p* = 0.036), with O in-between.

Regarding sugar entry into adipose tissue, the *glut4* glucose transporter was not modified by dietary intervention (Diet effect, *p* = 0.76) whereas the lactate *mct1* transporter expression was significantly increased in overfed animals regardless of fibre supplementation (+89% in O vs. C, *p* = 0.007; +134% in O + F vs. C, *p* < 0.001). Finally, *gpr43* (G protein-coupled receptor 43, specific for SCFAs) expression levels were down-regulated by overfeeding (−50% in O vs. C, *p* = 0.02), whereas fibre supplementation restored partially its expression, with no longer differences between the O + F and C groups. Concerning inflammation, *tnfα* expression was decreased after 2 months of overfeeding (−58% in O vs. C, *p* = 0.05) whereas the O + F group was similar to control normally fed animals. *Il-6* expression was not significantly modified.

At the gut level, gene expression of SCFA receptor *gpr41* in caecum, and pro glucagon, Glp1 and Glp2 encoding gene (*gcg*) in jejunum were both up-regulated in the fibre-supplemented group (+214% and +262% in O + F vs. O for *gpr41* (*p* = 0.01) and *gcg* (*p* = 0.03), respectively), as shown in Figure 2.

### 3.3. Fasting Concentration of Circulating Metabolites

Arterial concentrations of insulin, lactate, BHB, propionate, and selected AA (leucine, valine, isoleucine, phenylalanine, methionine, histidine, alanine, glutamate, glycine, serine, arginine, citrulline, cysteine, ornithine, asparagine, aspartate, taurine), as well as HOMA-IR, were significantly altered by overfeeding independently of fibre supplementation (Time effect, *p* < 0.5), but only alanine tended to be significantly different between the O and O + F groups (Diet effect, *p* = 0.07) (Table 2). Arterial values for part of these metabolites are already presented in a previous paper [13].

Many metabolites present at the hepatic vein level, thus representing those released by the splanchnic area and available for peripheral tissues, were significantly altered by overfeeding regardless of fibre supplementation, including lactate, BHB, acetate, total SCFAs and many AA including isoleucine, valine, phenylalanine, methionine, histidine, glutamate, glycine, arginine, citrulline, ornithine, asparagine and taurine (Time effect, *p* < 0.05) (Table 2). Remarkably, dietary fibre supplementation tended to increase hepatic vein concentrations for acetate and total SCFAs (Diet effect, *p* = 0.06), and glucose especially at D56 (Diet × Time, *p* = 0.07) (Table 2). Moreover, the appearance of other metabolites over time, such as alanine (Diet effect, *p* = 0.06, Diet × Time, *p* = 0.10), glutamine (Diet × Time effect, *p* = 0.08), and ornithine (Diet × Time effect, *p* = 0.07) tended to be altered specifically by dietary fibre supplementation.

### 3.4. Fecal SCFAs Concentration

Although SCFAs concentration in the artery was not altered by fibre supplementation, it was not the case within the lumen of the gut. Figure 3A shows the evolution of SCFAs concentrations in fecal water expressed as a percentage of the concentration measured at D1. During the experimental period (D14 and D56), total SCFAs in the feces (µmol/g feces) increased by 52 ± 16% in the F group, whereas it remained stable (+2.9 ± 16%) in the O group relative to the concentration of SCFAs at D0 (Diet effect, *p* = 0.05). Acetate content increase was more pronounced in the O + F than O group in both D14 and D56 (Diet effect, *p* = 0.06) (Figure 3A). Propionate and butyrate followed the same pattern of change (Diet effect, *p* = 0.05 and *p* = 0.08 for propionate and butyrate, respectively) (Figure 3A).

### 3.5. Fasting Nutrient Net Fluxes across the Splanchnic Area

The net portal drained viscera release of acetate (Diet effect, *p* = 0.11), and the net splanchnic release of acetate were not affected (Diet effect, *p* = 0.13) (Figure 3B). Contrary to acetate, a trend (Diet × time effect, *p* < 0.1) for an increased net release of propionate and butyrate in the portal vein of fibre-supplemented animals occurred at D14 (O + F vs. O, *p* = 0.04 and *p* = 0.03 for propionate and butyrate, respectively, post hoc analysis), but was no longer significant at D56 (Figure 3B). This occurred simultaneously with a numerically increased net hepatic uptake of propionate and butyrate at D14, even if not significant. At the overall splanchnic level, no significant differences on propionate and butyrate net release were found over the entire experimental period. In addition, the presence of these three major SCFAs in fecal water decreased over time independent of fibre supplementation (Time effect, *p* = 0.007, *p* = 0.002, *p* = 0.025 for acetate, propionate and butyrate, respectively), and similar patterns, although not always significant, were described for portal drained viscera, liver and total splanchnic fluxes, especially in the O + F group.

For glucose and lactate, the fibre supplementation did not impact on their net gut release in the portal vein (Figure 4). On the contrary, a significant Diet and Time × Diet effect for glucose (*p* < 0.05) and Diet effect (*p* = 0.06) for lactate were observed in the liver. Net hepatic release was increased by fibre supplementation at D56 for glucose (+320%, *p* = 0.01) and D14 for lactate (×17.9, *p* = 0.04). In addition, a Time effect (*p* = 0.001) within the fibre-supplemented group was also observed for glucose release. The consequence was an increased net release of glucose and lactate by the overall splanchnic tissues in the fibre- supplemented animals, especially at D56 for glucose (Diet × Time effect, *p* = 0.04) and at D14 and a lesser extent D56 for lactate (Diet effect, *p* = 0.005).

The fibre supplementation had no effect on net urea uptake (recycling) by the gut (net PDV uptake was similar between O and O + F), net hepatic release and net overall splanchnic release. For other metabolites (ammonia, BHB and individual AA), net PDV, hepatic and TSP fluxes were not strongly modified (no significant Diet or Diet × Time effects observed) by fibre supplementation (Appendix A).

## 4. Discussion

The aim of the present study was to explore, in the fasted state, the inter-organ cross-talk between gut, liver, muscle and adipose tissue on nutrients utilization and handling in order to explain the metabolic effects of fibre supplementation on peripheral tissues after adaptation to overnutrition in adult mini-pigs. Briefly, net splanchnic release of glucose and lactate increased when fibres were supplemented. In such situation, lipid entry and storage within the liver was also limited, as already published elsewhere [13]. This suggests that energy storage and utilization by the splanchnic area (and notably the liver) was decreased by fibre supplementation. Accordingly, we observed specific metabolic adaptations to this overall increased energy supply at the peripheral level, such as an increased capacity for fatty acid oxidation and mitochondrial activity in muscle and a raised fatty acids utilization and buffering capacity in adipose tissue, in fibre-supplemented mini-pigs.

### 4.1. Overfeeding and Fibre Supplementation: Impact on Muscle Structure and Metabolic Activities

Structural and phenotypic changes were observed at the muscle level (*Longissimus dorsi*) both after overnutrition and fibre supplementation. Of note, the increased average of muscle fibre CSA observed in animals from the O group is consistent with data already reported in obese humans [39,40,41] and Ossabaw pigs fed high fat/high cholesterol/high fructose [42] or Yucatan pigs fed a western diet [43]. This increased fibre CSA positively correlates with obesity and could be explained by a higher number of intramyocellular lipid droplets, as previously shown [42]. However, muscle triglycerides content remained unaltered after overfeeding in our study. The higher muscle fibre CSA in our O animals could be also due to a higher proportion of type II muscle fibres (lower size and glucose handling capacity) in detriment of highly-oxidative type I fibres, a feature associated with obesity in the literature [42,44]. Then, our data suggest that obesity-related alteration of muscle fibre CSA occurs before any impairments of lipid profile in the muscle.

This difference in muscle phenotype observed in our overfed fibre-supplemented pigs relative to overfed non-supplemented animals was also true metabolically as an increased capacity for fatty acids oxidation and an activation of mitochondrial activity was suggested by the gene overexpression of *acox*, *sdha*, *ucp2* and *cpt1-m* in O + F vs. O and C. This increased capacity for fatty acids catabolism and mitochondrial activity in the muscle, potentially leading to an increased utilization of lipids, can be one of the explanations why the liver remained relatively protected from an exacerbated lipids storage in dietary fibre-supplemented animals [13]. Such an up-regulation of muscle genes expression and enzymes involved in fatty acid oxidation and mitochondrial activity has been shown in rodents both after specific soluble fibres (i.e., *Psyllum* or epilactose) [22,45] or butyrate [21] supplementation when the animals were fed a high fat diet and developed obesity. In our case, the intake of a reasonable amount of a fibre-rich and ready-to-consume formulated bread was capable to switch muscle metabolism towards lipids utilization in an overfed mini-pigs model, suggesting that such a nutritional strategy could be transferable to humans. A mitochondrial dysfunction and a reduction of oxidative capacities in the muscle of obese and insulin-resistant volunteers/models have been repeatedly demonstrated [46,47], thus targeting mitochondrial dysfunction (as could be the case with dietary fibre supplementation) might improve muscle insulin signaling [48]. The pathway of lipid catabolism could also be fueled by a mobilization of fats within the muscle as suggested by an increased expression of the hydrolase *hsl* in fibre mix-supplemented animals, as already observed in muscle during exercise (i.e., when energy demand within the muscle is also stimulated) [49]. The existence of such a mechanism within the muscle of dietary fibre-supplemented animals, where muscle mitochondrial activity is also stimulated, requires to be further investigated. The triglyceride content within the muscle, similar within the three groups of animals cannot help to validate or invalidate this hypothesis.

A controller of energy metabolism/mitochondrial function in our study, *pgc-1α* significantly increased in the dietary fibre-supplemented group vs. O and C, and is a good candidate to modulate and stimulate energy catabolism. First, *pgc-1α* is known to be down-regulated during the development of insulin resistance, obesity and mitochondrial dysfunction [50,51] in parallel with *pgc-1α* target/associated genes: *nrf1*, *nrf2* or pparα [21,50,51]. Moreover, down-regulation of *pgc-1α* specifically in muscle favors chronic inflammation that, in turn, is closely associated to the production of reactive oxygen species (ROS) [52]. On the contrary, *pgc-1α* is generally up-regulated by butyrate or fibre supplementation [21,45,53] even if not always [54]. In the present study, together with the increased *pgc-1α* mRNA expression, associated genes *nrf2* and pparα [51] were significantly increased in muscle following fibre supplementation. Particularly, *nrf2* is known to respond to the increased ROS production by the activated mitochondria and oxidation processes [55]. In addition, transgenic mice overexpressing pparα in muscle exhibited increased fatty acid oxidation rates and were protected against diet-induced obesity [56]. Lastly, a link can be hypothesized between *pgc-1α* and fibre sizes: *pgc-1α* is known to be more expressed in muscles rich in type I fibres, which means that an enrichment in oxidative fibres in O + F animals can be hypothesized in our pigs’ muscles and explain the relatively smaller fibre area (lower average CSA) observed in O + F animals vs. O, since a higher oxidative capacity is generally associated with a lower fibre size in muscle [57].

In the present study, and contrary to what is generally found in the literature [58], the increased lipid oxidation does not seem to be associated with an increased stimulation of systems involved in lipids entry within the cell, as shown by the absent differences in the *slc27a4* (coding for FATP4) gene expression between O and O + F. However, lipid transporters, and among them *slc27a4*, are known to be regulated by complex translocation mechanisms [59], and not solely by transcriptional regulations. This means that an increased lipid entry within the cells following fibre supplementation in our situation of overfeeding cannot be entirely excluded, even if not measured.

### 4.2. Overfeeding and Fibre Supplementation: Impact on Adipose Tissue mRNA Levels

Such as for muscle, an increased lipids oxidation can be hypothesized in the subcutaneous adipose tissue of fibre-supplemented animals as suggested by an increased *cpt1* expression, the rate-limiting enzyme in mitochondrial fatty acid β-oxidation. In addition, the fact that *pparα* was also more stimulated in animals supplemented with the mix of fibres, sustains the hypothesis of an increased lipid catabolism in the adipose tissue. A stimulated fatty acids oxidation in adipose tissues has been reported in animal models following SCFAs or fibre supplementation [24,60], with a switch from lipogenesis to fat oxidation [24]. However, in our case, the impact of fibres on this pathway is not as straightforward because other genes expression (*acox*, *cpt2*, *sdha*, and *ucp2*) involved in mitochondrial activities or lipids catabolism are not differentially modulated in fibre-supplemented animals relative to O.

Lipid synthesis in the adipose tissues is stimulated by energy overload in overfed animals (supplemented or not with fibres), as shown by *fasn* gene expression up-regulation, thus suggesting that regulatory mechanisms of lipids handling following dietary fibre supplementation is more focused on the stimulation of catabolism than inhibition of synthesis. However, and contrarily to muscles, where an increased *hsl* gene expression in O + F animals tended to limit lipids storage as triglycerides and increase lipolysis within the muscle, no such specific regulatory mechanisms exist in adipose tissues as *hsl* mRNA levels remained unaltered whatever the diet considered. Concerning the lipid storage capacity via extracellular lipolysis mediated by lipoprotein lipase (*hsl*), the story is different because *angptl4* gene expression, an inhibitor of LPL, was down-regulated in the fibre- supplemented group. Therefore, an hypothesized increased LPL activity (due to a lack of inhibition of *angptl4*) in the O + F group might increase the buffering capacity of the adipose tissues by increasing LPL-mediated triglycerides extraction, clearance of triacylglycerol, and whole body insulin sensitivity [16], as also shown in *angptl4* KO mice [61]. In fact, an increased insulin sensitivity at the adipose level in fibre-supplemented animals can be expected as insulin is known to reduce *angptl4* gene expression in mice adipocytes [62]. This metabolic adaptation could prevent lipid deposition in other tissues, and notably liver, as already shown in a previous work [13]. As no alteration of plasma triglycerides or free fatty acids concentrations was observed throughout the experimental period of overfeeding, this hypothesis should require further investigation [13].

### 4.3. Consequences of the Activation of Lipid Catabolism in the Peripheral Organs on Overall Energy Metabolism

The metabolite β-OH-butyrate (BHB) can be synthesized from fatty acids (i.e., butyrate) [63] both in the gut or in the liver. At the hepatic level, we have already demonstrated an absence of impact of dietary fibre supplementation in overfed animals on fatty acid oxidation [13], therefore suggesting an unaltered hepatic BHB production rate. In line with these results, circulating levels of BHB were not altered by fibre supplementation. However, the increased butyrate levels present in the fecal water from fibre-supplemented animals suggested a potentially increased gut release of BHB to the periphery, as shown after butyrate infusion by Fitch et al. [63]. To explain this apparent discrepancy, the relative importance of BHB production at the gut level relative to BHB production from the beta-oxidation process in the liver to the final circulating BHB available to peripheral tissues should be much lower.

Nevertheless, the fibre-induced increase of lipid catabolism at the peripheral level might have generated an increased acetyl-CoA flux and pool size and the consequent activation of the Krebs cycle in muscle. As acetyl-CoA is known to inhibit pyruvate dehydrogenase kinase [64], accumulated pyruvate (from glycolysis) is redirected towards reduction into lactate and transamination into alanine in the muscle [65]. Therefore, the increased muscle lipids catabolism may induce a channeling of muscle acetyl-CoA towards Krebs cycle in the O + F animals, whereas pyruvate utilization is shifted from acetyl-CoA synthesis (from PDH activity) to alanine synthesis [66] and stimulation of the alanine-glucose cycle [67] at the whole body level. The tendency for an increased plasma concentration of alanine in artery and hepatic veins of overfed fibre-supplemented animals compared to overfed non-supplemented animals supports a preferred utilization of pyruvate towards synthesis of alanine in the muscle, whereas alanine is used for gluconeogenesis at the hepatic level (as observed with the increased net hepatic glucose production (Figure 2). Unfortunately, due to a high variability of the data, no significant increased alanine uptake was observed at the hepatic level to definitely validate this hypothesis of increased hepatic alanine utilization for glucose synthesis (Appendix A).

### 4.4. Mediators of These Metabolic Adaptation to Fibre Supplementation

Most of the studies have hypothesized the role of SCFAs as mediators of the beneficial effect of fibres on metabolism because in nearly all cases of fibre supplementation, there is an increase in fecal SCFA concentration [16,68]. Additionally, some of these studies reported an increased circulating SCFA content in the portal vein in animals supplemented with dietary fibres [69,70], with sometimes but not always an impact at the peripheral level as well [71]. Several authors have proven the role of fermentation products of fibres in the gut as mediators of the metabolic responses [16,21], with direct infusion/ingestion of SCFA in vivo [21] or in vitro on cell culture [72]. In the present study, we questioned the role of SCFAs as direct regulators of metabolic adaptations to overfeeding at the peripheral level as we failed to show significant alterations of net splanchnic emission of acetate, propionate and butyrate even if their concentrations were increased in fecal water. Part of this observation can arise from the fact that all measurements were made in the fasted state, whereas maximum production peak of SCFAs may arise within the first ten hours post meal ingestion, a factor that depends on the nature of fibres that reach the colon [73]. Aside from this “timing” effect, SCFAs produced from dietary fibres may have an indirect effect by inducing signals at the local level (i.e., gut level and hepatic level), as SCFAs receptors in the caecum (*gpr41*) and gut peptides synthesis in the jejunum (*gcg*) were up-regulated in fibre-supplemented animals compared to the O group. Remarkably, *gcg* gene encodes for a preprotein cleaved in four different peptides and among them GLP-1 and 2, which are involved in the regulation of gut growth and insulin sensitivity and secretion [74,75,76]. These peptides could take over the role of SCFAs at the peripheral level and should require further investigation as potential mechanisms of regulation of muscle/adipose tissue mitochondrial metabolic activity by dietary fibres. On the other hand, the time-dependent decrease of SCFAs concentrations reported in fecal water, independent of fibre supplementation, might be reflecting the potential deleterious impact of overfeeding to the gut microbiota composition and/or its capacity to generate these fermentation end-products. Hence, these results question the efficiency of fibre supplementation on health parameters on a long-term basis.

An increased sensitivity of peripheral tissues to similar amounts of SCFAs cannot be entirely excluded. Indeed, the overfeeding-induced decrease in GPR43 expression levels, a G-protein coupled receptor activated by SCFAs, was reverted by fibre supplementation in the adipose tissue. This is not the case in muscle, where the same gene is significantly down-regulated by overfeeding but not differentially expressed between the O and O + F groups (1.65 ± 0.67 vs. 1.50 ± 0.50, *p* = 0.695). Lastly, other molecules issued from catabolism of fibres within the caecum and colon and discussed in the literature could also be involved as regulatory molecules of energy metabolism and could improve insulin sensitivity of the host, such as succinate [77] or odd-chain fatty acids [78]. These potential effectors should be examined in detail, particularly via targeted (for succinate and odd-chain fatty acids) or untargeted (for still unknown molecules) metabolomics approaches, especially in the caecal water, portal plasma and tissues [79].

## 5. Conclusions

Our findings unveil several fibre-induced metabolic shifts involving nutrients and energy utilization in both the splanchnic area and peripheral tissues in a situation of overnutrition. Remarkably, we reveal an increased capacity for fatty acids catabolism and mitochondrial activity in the muscle of dietary fibre-supplemented mini-pigs that occurred in parallel with a shift of muscle fibre CSA into a lean-like profile. Then, we suggest that these features could be protecting the liver from lipids accumulation, although the undergoing mechanisms must be further evaluated, especially when an increased lipids entry within the myocytes could not be entirely excluded. In the adipose tissue, fibre supplementation did not impact as straightforward as for muscle. However, an increased *cpt1* expression, the rate-limiting enzyme in mitochondrial fatty acid β-oxidation, and a downregulated expression of LPL inhibitor *angptl4*, indicate a fibre-favored buffering capacity of adipose tissue by increasing triglycerides extraction and fatty acids utilization. Altogether, the fibre-induced increase of lipids catabolism at the peripheral level might be responsible for an increased alanine/glucose cycle, as supported by the higher levels of arterial plasma alanine and glucose net release by the liver. Among the main effectors of these metabolic adaptations, our data suggest that fibre-derived SCFAs are proposed to play a key role by locally signaling on up-regulated gastrointestinal targets (*gpr41* and *gcg*), thus highlighting the fibre consumption as a nutritional solution to improve insulin sensitivity by up-regulating GLPs incretin production. On the contrary, the SCFAs’ capacity to directly modulate energy storage and utilization at the peripheral level can be questioned. To conclude, the supplementation with a mix of dietary fibres favored the metabolic switch into lipid oxidation, especially in *Longissimus dorsi*, in overfed mini-pigs.

## Figures and Tables

**Figure 1 nutrients-13-04202-f001:**
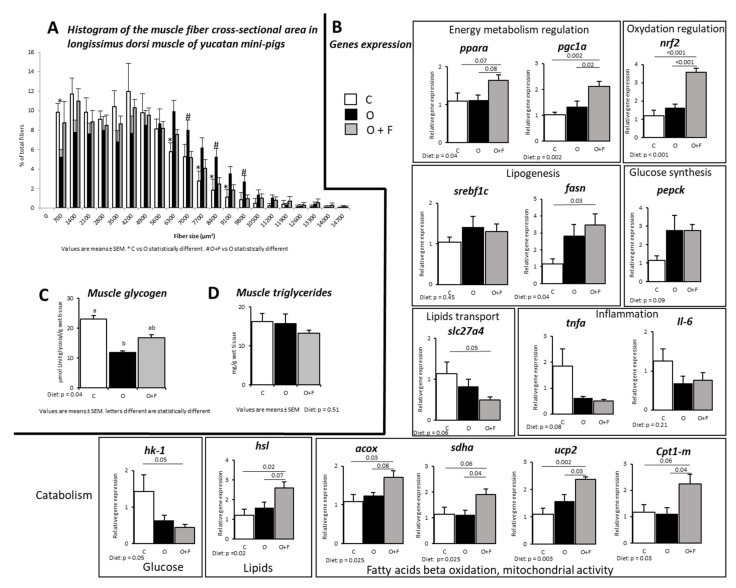
Muscle (longissimus dorsi) metabolic activity in mini-pigs fed a control (C) diet, following 56 days of overnutrition (O) or 56 days of overnutrition and supplementation with a mix of dietary fibres (O + F). (**A**) Histogram of the muscle fibres cross-sectional area. (**B**) Gene expression in the fasted state. (**C**) Fasting glycogen content in muscle at D56. (**D**) Fasting triglycerides content in muscle at D56. Data are expressed as the mean ± S.E.M. Significant differences are considered when *p* < 0.05. *acox*, Acyl-CoA oxidase; *cpt1-m*, carnitine O-palmitoyltransferase 1 muscular; *fasn*, fatty acid synthase; *hk-1*, hexokinase-1; *hsl*, hormone-sensitive lipase; *Il-6*, interleukine-6; *nrf2*, nuclear factor erythroid 2-related factor 2; *pepck*, phosphoenolpyruvate carboxykinase; PGC-1α, peroxisome proliferator-activated receptor gamma coactivator 1-alpha; *pparα*, peroxisome proliferator-activated receptor alpha; *sdha*, succinate dehydrogenase complex flavoprotein subunit A; *slc27a4*, solute carrier family 27 member 4; *srebp1c*, sterol regulatory element-binding protein; *tnfα*, tumor necrosis factor alpha; *ucp2*, uncoupling protein 2.

**Figure 2 nutrients-13-04202-f002:**
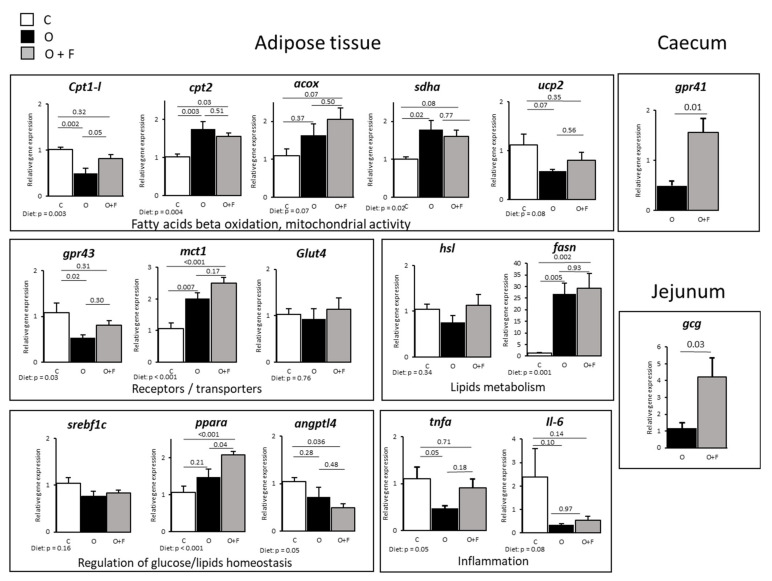
Adipose (subcutaneous) tissue, caecum and jejunum genes expression in mini-pigs fed a control (C) diet, following 56 days of overnutrition (O) or 56 days of overnutrition and supplementation with a mix of dietary fibres (O + F). Data are expressed as the mean ± S.E.M. Significant differences are considered when *p* < 0.05. *acox*, acyl-CoA oxidase; *angptl4*, angiopoietin-like 4; *cpt1* and *2*, carnitine O-palmitoyltransferase 1 and 2; *fasn*, fatty acid synthase; *gcg*, glucagon precursor; *glut4*, glucose transporter type 4; *gpr43* and *41*, G-protein-coupled receptor 43 and 41; *hsl*, hormone-sensitive lipase; *Il-6*, interleukine-6; *mct1*, monocarboxylate transporter 1; *pparα*, peroxisome proliferator-activated receptor alpha; *sdha*, succinate dehydrogenase complex flavoprotein subunit A; *srebf1c*, sterol regulatory element-binding protein; *tnfα*, tumor necrosis factor alpha; *ucp2*, uncoupling protein 2.

**Figure 3 nutrients-13-04202-f003:**
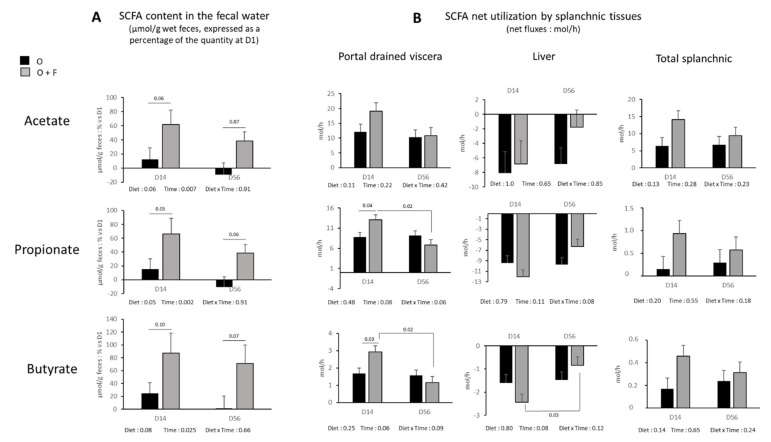
Short chain fatty acids (SCFA) (acetate, propionate and butyrate). (**A**) Concentration in the fecal water (µmol/g wet feces) expressed as a percentage of the quantity at the beginning of the experimental period (D1). (**B**) Net flux across the splanchnic tissues (mol/h) (portal drained viscera (PDV), liver, and total splanchnic tissues (TSP: PDV + liver) after 14 (D14) and 56 (D56) days of overnutrition (O) or overnutrition and supplementation with a mix of dietary fibres (O + F). Data are expressed as the mean ± S.E.M. Significant differences are considered when *p* < 0.05.

**Figure 4 nutrients-13-04202-f004:**
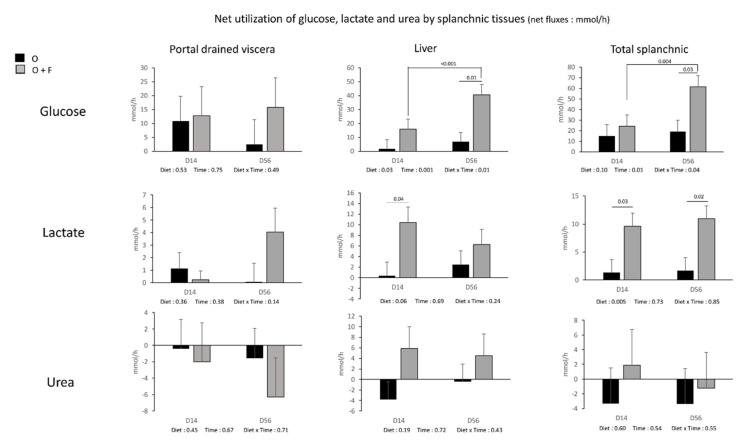
Glucose, lactate and urea net fluxes (mmol/h) (across the splanchnic tissues (portal drained viscera (PDV), liver, and total splanchnic tissues (PDV + liver) TSP)) after 14 (D14) and 56 (D56) days of overnutrition (O) or overnutrition and supplementation with a mix of dietary fibres (O + F). Data are expressed as the mean ± S.E.M. Significant differences are considered when *p* < 0.05.

**Table 1 nutrients-13-04202-t001:** Primer sequences used for qrt-pcr.

Gene Symbol	Primer Sequence (Forward)	Primer Sequence (Reverse)
*Acox*	GACCTGAGCGGCCTACCTGA	CATCAGGAACCTGGCCGTCT
β-*Act*	GCGGCATCCACGAAACTACC	CTCTGGAGGCGCGATGATCT
*Angptl4*	CTCTGGTGGTTGGTGGTTTG	GCTACTGTGGGCTGGATCA
*Cpt1*	GACGAGGACCCCTGATGGTG	CGTGGATCCCAGGAGAATCG
*Cpt-m*	CTGCAGGGGGAAGAGTGGAG	TTGAACGCGATGAGGGTGAA
*Cpt2*	TACGAGTCCTGTAGCACTGC	TGGTTGTGGTACTCGGAACA
*Eef1a*	ACCTGTGCTGGATTGCCACA	AACAGCAAATCGGCCCAGAG
*Fasn*	CGGTTCCAAGGAGCAAGGTG	GCATTCACGATGCCGTTCAG
*Gapdh*	ACGGTCCATGCCATCACTG	CCAGTGAGCTTCCCGTTGA
*Gcg*	CCCAGGATTTTGTGCAGTGG	GCAATGAATTCCTTGGCAGC
*Gpr41*	CTCCGTGTACCTCTTGACGT	AGACGGTGGTGAAGAAGAGG
*Gpr43*	GAGTGATTGCTGCTCTGGTG	TGGGGATGAAGAAGAGGACG
*Glut4*	GCCCGCGAGAAAGAGTCTGA	GCCGTCTCGAAGATGCTGGT
*Hk1*	AGCTAAGAGTCCTGGCCCCC	CGCCATTAGGTGGCTTCTGC
*Hprt1*	TATGGACAGGACTGAACGGC	TGGTCATTACAGTAGCTCTTCAG
*Hsl*	CGTCTCTAGCAAACATGGCA	TCACTGTCCTGTCCTTCACG
*Il-6*	TGGGTTCAATCAGGAGACCT	GTGGTGGCTTTGTCTGGATT
*Mct1*	TGGCTGTCATGTATGGTGGA	AAGCCCAAGACCTCCAATGA
*Nrf2*	ATTCCCAGGTTTCTTCGGCT	TGGAACCGTGCTAGTCTCAG
*Pepck-m*	CTGGAAACCCGGTGACAAGG	GGGGGACTCCTTTGGGTCTG
*PGC-1α*	AGGCAGAAGGCAATTGAAGA	TTTCAAGAGCAGCAAAAGCA
*Pparα*	CAGCAATAACCCGCCTTTCG	ACTTGGCGAACTCCGTGAGC
*Pparg*	TGTGAAGGATGCAAGGGTTTC	CAACAGCTTCTCCTTCTCAGC
*Rps9*	TTGAAGGGAATGCTCTGCTG	GGACAATGAAAGGACGGGATG
*Srebp-1c*	AGGCAGCACCTTTGCAGACC	GCGGTAGCGTTTCTCGATGG
*Sdha*	CCTCCGTGGTAGAGCTAGAG	TACCGCAGAGACCTTTCCGTA
*Slc27a4*	TGCAGTACATTGGCGAGCT	ACACTGGCCGTCAAAGTTG
*Tbp*	TTTTGGAGGAGCAGCAAAGG	GTGGAAGAGCTGTGGAGTCT
*Tnfa*	TGTAGCCAATGTCAAAGCCG	ATGGCAGAGAGGAGGTTGAC
*Ucp2*	TCGACGCCTACAAGACCATC	GCAGGGAAGGTCATCTGTCA

Acox, Acyl-coA oxidase 1 and 2; β-Act, β-actin; Angptl4, angiopoietin like 4, Cpt1, Carnitine o-palmitoyl transferase 1; Cpt1-m, Carnitine o-palmitoyl transferase 1 muscular; Cpt2, Carnitine o-palmitoyl transferase 2; Eef1a, Elongation factor 1-alpha; Fas, Fatty acid synthase; Gapdh, Glyceraldehyde-3-phosphate dehydrogenase; Gcg, Glucagon precursor; Gpr41, G protein-coupled receptor 41; Gpr43, G protein-coupled receptor 43; Glut4, Glucose transporter type 4; Hk1, Hexokinase-1; Hprt1, Hypoxanthine phosphoribosyltransferase 1; Hsl, Hormone-sensitive lipase; Il-6, Interleukin-6; Mct1, Monocarboxylate transporter 1; Nrf2, Nuclear factor erythroid 2–related factor 2; Pgc-1α, Peroxisome proliferator-activated receptor-gamma coactivator-1α; Pepck-m, Phosphoenolpyruvate carboxykinase muscular; Pparα, Peroxisome proliferator-activated receptor alpha; Pparg, Peroxisome proliferator-activated receptor gamma; Rps9, Ribosomal protein S9; Srebp-1c, Sterol regulatory element-binding protein; Sdha, Succinate dehydrogenase complex flavoprotein subunit A; Slc27a4, Solute carrier family 27 member 4; Tbp, TATA-box binding protein; Tnfα, Tumor necrosis factor; Ucp2, Uncoupling protein 2.

**Table 2 nutrients-13-04202-t002:** Plasma biochemical parameters in mini-pigs in the fasted state before (D1) and after 14 (D14) and 56 days (D56) of adaptation to overfeeding and control bread supplementation (O) or overfeeding and bread supplemented with fibre mixture (O + F) in the artery (A) or hepatic vein (HV).

		O				O + F				*p* Value		
		D1	D14	D56	SEM	D1	D14	D56	SEM	Diet	Time	Diet × Time
Insulin (µg/L)	A	0.05	0.15	0.14	0.02	0.09	0.13	0.14	0.02	0.68	0.03	0.54
HOMA-IR (AU)	A	0.26	0.90	0.82	0.12	0.55	0.75	0.85	0.12	0.75	0.04	0.44
Metabolites (mmol/L)												
Glucose	A	4.93	5.57	5.26	0.14	5.22	5.15	5.34	0.14	0.93	0.28	0.16
	HV	5.81	5.93	5.74	0.38	5.74	5.71	6.57	0.33	0.72	0.22	0.07
Lactate	A	0.42	0.62	0.68	0.02	0.44	0.58	0.72	0.02	0.76	<0.001	0.70
	HV	0.50	0.64	0.74	0.05	0.44	0.80	0.94	0.04	0.14	<0.001	0.17
Urea	A	4.30	3.78	3.94	0.17	4.15	3.60	3.56	0.18	0.36	0.10	0.90
	HV	4.40	3.74	4.14	0.24	4.23	3.66	3.53	0.21	0.39	0.16	0.67
Ammonia	A	42.1	44.0	42.2	5.6	36.3	37.7	45.0	5.6	0.70	0.65	0.58
	HV	61.7	52.9	58.6	7.9	46.3	39.4	52.6	6.9	0.28	0.21	0.70
β-hydroxybutyrate	A	11.5	8.1	8.8	0.8	14.8	7.2	7.3	0.8	0.80	0.01	0.32
	HV	15.0	10.8	11.4	2.3	24.5	10.6	13.3	1.9	0.24	0.03	0.33
Acetate	A	613	755	679	15	664	658	664	15	0.37	0.11	0.08
	HV	692	886	786	32	819	931	880	28	0.06	0.04	0.76
Propionate	A	20.7	27.4	22.1	0.6	22.6	23.3	20.3	0.6	0.13	0.01	0.11
	HV	27.3	31.1	27.7	3.3	25.5	36.7	33.1	2.9	0.50	0.39	0.75
Butyrate	A	2.97	5.45	5.40	0.75	3.15	5.01	2.53	0.75	0.34	0.13	0.37
	HV	8.53	9.04	7.69	1.72	8.15	12.8	9.19	1.50	0.48	0.14	0.36
Σ SCFA	A	636	788	707	16	689	687	687	16	0.34	0.09	0.08
	HV	728	926	822	34	853	981	922	29	0.06	0.04	0.82
Amino acids (mmol/L)												
Leucine	A	177	193	211	8	171	183	211	9	0.65	0.01	0.90
	HV	172	200	224	9	176	196	214	8	0.76	0.03	0.91
Isoleucine	A	126	157	144	7	119	157	147	7	0.90	0.004	0.88
	HV	126	162	148	8	121	165	149	7	0.95	0.01	0.94
Valine	A	324	314	362	11	314	313	365	12	0.85	0.04	0.94
	HV	308	314	386	9	317	317	378	8	0.97	0.04	0.91
Lysine	A	166	165	156	8	163	163	177	8	0.62	0.96	0.39
	HV	166	168	157	9	155	163	172	8	0.95	0.91	0.57
Phenylalanine	A	64.1	71.3	77.2	3.4	63.4	65.3	76.7	3.5	0.63	0.005	0.66
	HV	59.1	69.1	76.3	3.6	61.7	65.2	72.1	3.2	0.70	0.002	0.50
Methionine	A	22.4	26.8	25.2	1.9	22.9	29.4	25.1	2.0	0.72	0.03	0.77
	HV	21.3	26.4	21.9	1.8	22.1	30.2	25.0	1.6	0.31	0.004	0.70
Threonine	A	156	161	166	8	164	164	172	9	0.65	0.57	0.94
	HV	142	156	167	7	158	156	167	6	0.59	0.23	0.65
Tryptophane	A	51.6	53.9	45.8	3.1	49.2	58.3	49.5	3.2	0.68	0.17	0.71
	HV	52.5	59.3	49.0	4.6	51.4	56.8	48.4	4.1	0.83	0.17	0.98
Histidine	A	90.0	101.6	104.3	4.6	91.8	100.6	108.0	4.7	0.82	<0.001	0.76
	HV	90.7	100.9	100.0	4.8	90.6	102.1	108.1	4.3	0.64	<0.001	0.33
Alanine	A	210	280 *a*	238 *a*	12	206	325 *b*(*t*)	296 *b*	12	0.07	<0.001	0.22
	HV	190	240 *a*	192 *a*	16	172	309 *b*	270 *b*	14	0.06	0.003	0.10
Glutamate	A	127	182	154	12	114	167	189	13	0.90	<0.001	0.17
	HV	218	321	358	52	221	357	402	46	0.69	<0.001	0.79
Glutamine	A	357	375	373	22	378	400	390	23	0.52	0.44	0.97
	HV	329	363	277	24	324	329	328	22	0.90	0.07	0.08
Glycine	A	801	982	849	38	865	1036	880	39	0.38	0.002	0.94
	HV	819	1008	823	53	852	994	885	47	0.71	0.03	0.82
Proline	A	308	338	297	8	295	347	325	8	0.49	0.13	0.63
	HV	308	316	299	16	292	333	361	14	0.36	0.52	0.37
Tyrosine	A	82.9	89.0	86.4	8.0	78.4	90.3	87.6	8.2	0.96	0.18	0.79
	HV	79.0	87.5	88.3	9.8	72.3	86.7	79.5	8.7	0.68	0.12	0.71
Serine	A	128	176	164	7	144	186	177	7	0.20	<0.001	0.94
	HV	113*a*	168	156	7	145*b*	175	167	6	0.10	<0.001	0.16
Arginine	A	107	126	126	4	111	126	140	4	0.29	0.003	0.51
	HV	106	128	122	5	109	124	151	4	0.17	0.01	0.12
Citrulline	A	82.8	85.0 *a*	86.2	4.1	77.9	103.1 *b*(*t*)	97.5	4.6	0.20	0.03	0.09
	HV	94.3	109.4	104.8	6.8	89.5	127.6	116.3	6.4	0.36	<0.001	0.11
Cystine	A	75.0	54.8	56.2	2.0	72.8	59.5	64.0	2.1	0.26	0.002	0.54
	HV	80.5	51.9	57.5	2.2	75.7	61.4	63.5	2.0	0.27	0.008	0.51
Ornithine	A	72.4	95.6	69.4 *a*	2.4	68.8	90.7 *b*(*t*)	80.2	2.5	0.84	<0.001	0.10
	HV	83.9	98.2	74.5	3.4	72.7	94.2	85.0	3.0	0.74	<0.001	0.07
Asparagine	A	18.7	35.6	21.1	2.9	19.4	30.2	23.5	2.9	0.86	0.003	0.56
	HV	18.1	34.3	17.2	4.8	7.9	16.8	12.1	4.3	0.11	0.03	0.39
Aspartate	A	8.6	12.4	8.4	0.97	7.3	12.1	9.5	1.0	0.91	0.002	0.62
	HV	10.0	13.2	10.4	1.3	9.8	15.9	12.8	1.2	0.38	0.01	0.55
Taurine	PV	104	155 *a*	117	9	95	116	108b	10	0.15	0.001	0.10
	HV	109	136	108	8	106	118	106	7	0.44	0.002	0.25
LBP (ng/L)	A	3.16	2.94	3.68	0.43	2.51	1.71	2.99	0.43	0.18	0.25	0.85
	PV	3.07	3.02 *a*	4.24	0.48	1.46	0.93 *b*(*t*)	3.00	0.51	0.03	0.11	0.86
	VSH	3.18	3.03	3.35	0.48	2.51	1.78	3.17	0.40	0.28	0.32	0.63
IL6 (pg/mL)	A	5.09	4.90	5.45	0.45	4.23	4.29	5.95	0.46	0.62	0.20	0.57

Values are means *p*-values and were obtained by Two Way Repeated Measures ANOVA. For a metabolite, within the same time point, O and O + F are significantly different (*a*,*b*) or tend (*t*) to be different.

## Data Availability

The data presented in this study are available on request from the corresponding author.

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
