# Peer review of "A Mix of Dietary Fibres Changes Interorgan Nutrients Exchanges and Muscle-Adipose Energy Handling in Overfed Mini-Pigs"

_nutrients, 2021, doi:10.3390/nu13124202_

Round 1
Reviewer 1 Report
The authors have a nice minipig model for these studies that allows determination of metabolite fluxes as well as large amounts of tissue for analysis. This study is a secondary analysis of data that uses the same pigs as a previously published work. This work has been cited appropriately.
The main issue with this manuscript is the use of "overfeeding" to describe the model, when there is no evidence either in this paper or the original report (ref 13) of caloric intake to show that the pigs did indeed overeat. Even in Ref 13, it is unclear whether the weight gain of the O group is in excess of any control, since none was included in that paper. Thus, there is actually no concrete evidence of over-feeding. Perhaps the differences were actually caused by the altered macronutrient composition of the diet or reduced intake of essential elements such as choline, which is known to affect body weight. Adding sugar and starch to the standard diet dilutes out essential nutrients like vitamins and minerals; this may also may affect intake of essential fatty acids and protein. Since nutritional profiles of the diets and nutritional intake data were not included in Ref 13, it would be appropriate and helpful to include such information here.
This is also an issue in the literature review, where vague terms like “over-fed” should be replaced by specification of the diet or how caloric excess was achieved. Most of the conditions mentioned in refs 3-6 involve some degree of overnutrition e.g. obesity, diabetes, CVD, cancers.
The results describing gene expression in muscle and adipose tissue are complex and described in a rather convoluted way. It seems there are 4 patterns: genes altered by fat, genes altered by fibre, genes altered by fat and “normalized” by fibre, genes that don’t change. Perhaps there is a way to use these categories to more simply describe the results.
The gene names are appropriately no caps, italicized in the figures but CAPS in the text and figure legends. Also regarding the figures, for simplicity, it is suggested to only note significant differences with bars and p-values. Noting all the trends on the figures, tables and in the text gives them more importance than warranted.
Figure 3A – for the Y-axis, do the authors mean “% change from D1”? As currently indicated, it appears that SCFA were all lower than baseline.
As the authors discuss (paragraph beginning line 458), an obvious line of investigation to support enhanced insulin signaling could be presumably be investigated using tissues at hand and would increase the impact of this manuscript if it were included here.
Minor points
Line 29 – GLP1 (from gcg) is an incretin, not a digestive peptide
Line 137 and 139 – change “indispensable” to “essential” and non-indispensable to non-essential
Delete row 178
Line 182 – RNAse-free column
Line 248 – unclear what “sides” are being referenced
Line 364 – the phrase “emitted by splanchnic area” has unclear meaning
Reviewer 2 Report
Dear colleagues,
thank you very much for the opportunity to review this extraordinary interesting work.
This topic is very complex, but your study is well- thought- out and planed, so is gives a perfect overview on the complicated network "gut- liver- muscle- adipose tissue" and comes to a conclusion.
- The article is well structured, the figures and tables are complex, but necessary to understand the results. Only the supplementary table is mentioned in the discussion for the very fist time- this occurs illogical. Please remove this S- table up to the results section and introduce it there! (Line 568- 571)
- The topic is very complex. The introduction is short and clear, the aim of the study is clearly worded. Both results section and discussion are sub classified in question related sections, witch makes the article more understandable.
- I am missing a short explanation why you took the special Yucatan pigs to approach this metabolic network. Please mention advantages to other animal models, known from the literature.
- I am also missing some correlation of this complex metabolic network behavior to the potential human findings on this field. In short segments probably in the introduction, as well as at the end of the discussion part (transition into a clinical understanding of obesity in human)
- The sentence in line 214, mentioning Reference (29) occurs too straight. Please build a short statement. Reference remains in brackets.
- Same with line 241 and reference (13)
- Please give a short statement in the results section to the reason of taking measurements on day 1, 14 and 56... is there some specific reason on that (previous studies)
- You are mentioning the weight of the animals at the beginning of the experiment, but no more at later times. How is the weight changing? Is it changing? And does the weight have any relation to the intake of fibre modified diet or is it similar in the groups C, O and O-F?
Overall a very interesting paper to topic of high complexity. It should be published after minor corrections.
Round 2
Reviewer 1 Report
No further comments
This manuscript is a resubmission of an earlier submission. The following is a list of the peer review reports and author responses from that submission.